# A Short-Term Wind Speed Forecasting Model Based on EMD/CEEMD and ARIMA-SVM Algorithms

**Ning Chen** [1,2,*], **Hongxin Sun** [1], **Qi Zhang** [1] and **Shouke Li** [1,2]

[1] School of Civil Engineering, Hunan University of Science and Technology, Xiangtan 411201, China; cehxsun@hnust.edu.cn (H.S.); 19020201048@mail.hnust.edu.cn (Q.Z.); lishouke@hnust.edu.cn (S.L.)
[2] Hunan Provincial Key Laboratory of Structures for Wind Resistance and Vibration Control, Hunan University of Science and Technology, Xiangtan 411201, China
[*] Correspondence: xningchen@hnust.edu.cn

**Abstract:** In order to ensure the driving safety of vehicles in windy environments, a wind monitoring and warning system is widely used, in which a wind speed prediction algorithm with better stability and sufficient accuracy is one of the key factors to ensure the smooth operation of the system. In this paper, a novel short-term wind speed forecasting model, combining complementary ensemble empirical mode decomposition (CEEMD), auto-regressive integrated moving average (ARIMA), and support vector machine (SVM) technology, is proposed. Firstly, EMD and CEEMD are used to decompose the measured wind speed sequence into a finite number of intrinsic mode functions (IMFs) and a decomposed residual. Each of the IMF subseries has better linear characteristics. The ARIMA algorithm is adopted to predict each of the subseries. Then, a new subseries is reconstructed using the sum of the predicted errors of all subseries. The high nonlinear features of the reconstructed error subseries are modeled using SVM, which is suitable to process nonlinear data. Finally, the superposition of all prediction results is performed to obtain the final predicted wind speed. To verify the stability and accuracy of the model, two typhoon datasets, measured from the south coast of China, are used to test the proposed methods. The results show that the proposed hybrid model has a better predictive ability than single models and other combined models. The root mean squared errors (RMSEs) of the hybrid model for the three wind speed datasets are 0.839, 0.529, and 0.377, respectively. The combination of CEEMD with ARIMA contributes most of the prediction performance to the hybrid model. It is feasible to apply the hybrid model to wind speed prediction.

**Keywords:** wind speed prediction; empirical model decomposition; autoregressive integrated moving average; support vector machine; hybrid model

## 1. Introduction

Gales are one of the main causes of ground transportation meteorological disasters, especially in the windy areas of southeast coastal China. On typhoon days, severe convective weather, accompanied by strong winds and heavy rains, seriously threatens the driving safety of high-speed trains and road vehicles. In order to ensure the driving safety of vehicles, the wind monitoring and warning system [1] is becoming one of the major mitigation measures for wind-induced transportation accidents. As the monitoring value, or the forecasted wind speed, exceed the alarm threshold, the early-warning signal is issued to alert the running vehicles, avoiding wind-induced accidents [2,3]. Thereby a highly accurate short-time wind speed forecasting model is particular important to ensuring reliable operation of the system.

In general, wind speed prediction methods can be roughly divided into two categories according to the prediction principle [4]: (1) physical meteorological models based on atmospheric dynamic equations; (2) statistical methods based on historical observation

data. The statistical methods are constructed based on vast historical observations of wind speed. These are particularly suitable for short-term wind speed forecasting. Conventional statistical models, such as autoregressive models [5], neural network models [6,7], Kalman filters [8], and support vector machines [9] are currently widely used. However, the wind has the characteristics of randomness, periodicity, non-stationarity, and nonlinearity. In order to further improve the accuracy and stability of the wind speed prediction algorithm, the hybrid methods, combining the complimentary features and advantages of various methods, have been favored by scholars.

The hybrid wind speed forecasting methods based on decomposition techniques are developing rapidly. Techniques such as EMD (Empirical Mode Decomposition) and WPT (Wavelet Packet Transform) [10], are used as data preprocessors to decompose wind series or eliminate stochastic volatility. The fast ensemble EMD is adopted by Hui Liu et al. [11] to decompose the original wind speed series into a number of sub-layers, and the MLP (Multi-Layer Perceptron) neural networks optimized by MEA (Mind Evolutionary Algorithm) and GA (Genetic Algorithm) are built to predict the decomposed wind speed sub-layers. Two hybrid methods are proposed for the accurate multi-step wind speed prediction. Chi Zhang et al. [12] develop an EMD-based decomposition selection forecasting (DSF) model for wind speed prediction. The DSF model is achieved using artificial neural networks (ANNs) and support vector machines (SVM), which is verified to be effective with great precision. Jianzhou Wang et al. [13] propose a hybrid forecasting approach that combines the extreme learning machine, the Ljung-Box Q-test, and the seasonal ARIMA to enhance the accuracy of wind speed forecasting; the results show that the developed hybrid method exhibits stronger forecasting ability. K. R. Nair et al. [14] investigate the prediction performances of the hybrid model combining ARIMA and ANN. Madasthu Santhosh et al.[15] construct a hybrid wind speed prediction model integrating ensemble EMD and an adaptive wavelet neural network (AWNN), which has high accuracy, strong stability, and advantages of a small amount of calculation. A big hybrid WPD-Boost-ENN-WPF framework for multi-step wind speed prediction, consisting of wavelet packet decomposition (WPD), Elman neural network (ENN), boosting algorithms, and wavelet packet filter (WPF) are proposed [16]. On the basis of a hybrid model decomposition method (HMD) and online sequence outlier robust extreme learning machine (OSORELM), a hybrid short-term wind speed prediction model is developed by Zhang Dan et al. [17]. The results show that the online model and deep decomposition technology improve the stability and accuracy of model prediction. Considering the correlation of the prediction errors, Xu Yuanyuan et al. [18] propose an EMD-SVM model, with error compensation to reduce the accumulated errors and improve the prediction accuracy of short-term wind speed forecasting. Tianyu Tao et al. [19] present a performance evaluation of linear and nonlinear models for the short-term forecasting of tropical storms. Liu Mingde et al. [20] propose a hybrid EMD-RNNs-ARIMA model for wind speed and wind power prediction. Li Zheng et al. [21] combine the improved sparrow search algorithm (ISSA) and least squares SVM (LSSVM) to improve the convergence accuracy and shorten the prediction time of the wind prediction model. Hu Haize et al. [22] introduce a gray wolf algorithm (GWO) and SVM wind prediction model to predict the wind speed accurately.

Therefore, it is clear to see that the hot point of the wind speed forecasting models is mainly aiming at the development of hybrid models. A hybrid model has the advantage of higher accuracy, better stability, and greater adaptability. Mode decomposition technology, such as CEEMD and EMD, is an effective algorithm for processing nonlinear and non-stationary signals, which has been widely used in the wind speed forecasting model. It favors reducing the influence of unfavorable factors such as randomness and volatility of the original wind speed on the prediction errors. In this paper, hybrid short-term wind speed forecasting models combined with CEEMD/EMD-ARIMA-SVM are proposed. The linearity of ARIMA and the globe nonlinearity of SVM are fully exploited in the algorithm to improve the accuracy of the forecasting model. The validity of the model is verified

based on the measured typhoon wind speed data; results show that the proposed prediction model has sufficient accuracy and great stability.

## 2. Methodology

### 2.1. Modal Decomposition Technique

In order to effectively deal with nonlinear and non-stationary data, Huang. E.N. et al. [23,24] proposed the empirical mode decomposition (EMD) method.

EMD decomposes the original time series into a finite number of intrinsic mode functions (IMFs) and a residual component. IMFs must satisfy two conditions: (1) The number of the extremums and zeros of all IMF datasets must be the same or with a maximal difference of one; and (2) The average value of the upper and lower envelopes composed of the local maximums and local minimums at any point is zero.

The IMF extracts the local variation characteristics of the signal, which reflects the internal change of the signal. Therefore, the EMD method has a strong adaptability to the decomposing signal. It actually decomposes the signal into different scales of fluctuations or trend terms step by step, which eliminates the influence between the components. Therefore, the impact of the non-stationarity of signals could be diminished.

In terms of time series $y(t)$, it rewrites it as the following expression, processed by EMD. It gets:

$$y(t) = \sum_{j=1}^{m} c_j(t) + r(t) \tag{1}$$

where $c_j(t)$ is the *j*th IMF component; it represents the *j*th separated signal component from the original signals, which present different characteristic scales of the signals; $r(t)$ is the residual component reflecting the trend term of the original signal.

However, one of the major drawbacks of the original EMD is the frequent appearance of modal mixing, which makes the physical meaning of individual IMF unclear. In order to relieve the influence of the mode mixing, the complementary ensemble empirical mode decomposition (CEEMD) is proposed by Yeh, J.R. et al. [25]. The CEEMD is achieved through the following steps: firstly, the original time series is reconstructed by adding a series of pairs of Gaussian white noises with equal size and opposite direction; subsequently, with the applying of the EMD to each pair of the reconstructed time series, the final IMFs are obtained by means of averaging all of the EMD results of the IMFs. On the one hand, this improved method suppresses the defect of modal mixing of the original EMD algorithm; on the other hand, the auxiliary white noise signals will produce a mutual cancellation effect after decomposition and superposition, and the original signal will not be greatly affected by the addition of white noise. The detailed steps of the CEEMD algorithm are as follows:

1. The original signal $y(t)$ is added with a pair of Gaussian white noises $\varepsilon_j(t)$ to form a new set of signals $G(t)$, namely

$$\begin{aligned} G_1(t) &= y(t) + \varepsilon_i(t) \\ G_2(t) &= y(t) - \varepsilon_i(t) \end{aligned} \tag{2}$$

2. The EMD is applied to the reconstructed signal of Equation (2) to obtain m IMF components:

$$G_1(t) = \sum_{j=1}^{m} C_{ij,1}(t) + r_{i,1}(t)$$

$$G_2(t) = \sum_{j=1}^{m} C_{ij,2}(t) + r_{i,2}(t)$$

(3)

where, $G_{ij,\cdot}(t)$ indicates the $j$th IMF of the EMD decomposition after the adding of the $i$th white noise; $r_{i,\cdot}(t)$ represents the trend term of the EMD decomposition.

3. Different white noises $\varepsilon_i(t)$ ($i = 1, 2, \ldots, n$), are added, repeating steps 1 and 2, getting $n$ sets of IMFs and trend terms.
4. The mean of all IMFs are calculated to obtain the final IMF $c_j(t)$:

$$c_j(t) = \frac{1}{2n} \sum_{i=1}^{n} \left( C_{ij,1}(t) + C_{ij,2}(t) \right) \quad (j = 1, 2, \cdots, m)$$

(4)

### 2.2. The Auto-Regressive Integrated Moving Average Models

Wind speed prediction can be treated as a time series prediction problem. The autoregressive integrated moving average model (ARIMA) [5] has been one of the most popular approaches to forecasting, as it is robust and easy to implement. The ARIMA model is comprised by three parts, i.e., the autoregressive (AR) model, moving average (MA) model, and an integrated part (I) achieved using the differential. In term of the non-stationary data, the ARIMA model preprocess the data using the differential to make the data stationary. Thus, the ARIMA model is suitable for the forecasting of the non-stationary time series. After that, the ARIMA model treats the future value of the prediction as a linear combination of the past observations and pure random errors. The ARIMA ($p$, $q$, $d$) forecasting model for time series $y_t$ can be expressed as follows [26,27]

$$\Phi(B)\nabla^d (y_t - \mu) = \Theta(B)\varepsilon_t, \ t = 1,2,\ldots,T$$

(5)

where $y_t$ is the time series at time period $t$; $\mu$ is a constant, representing the mean value of the time series; $\varepsilon_t$ is the random errors at time period $t$; supposing that the random errors are independent and identically distributed, with a mean of zero and a constant variance, i.e., $E(\varepsilon_t) = 0$ , $Var(\varepsilon_t) = \sigma^2$ ; $B$ is the backward shift operator, $y_{t-p} = B^p y_t, \forall p \geq 1$; $d$ represents the order of difference; $\nabla$ is difference operator, $\nabla^d = (1 - B)^d$; $\Phi(B)$ is the polynomial of AR model, $\Phi(B) = 1 - \sum_{i=1}^{p} \phi_i B^i$ , $p$, $\phi_i$ are the order and coefficient of the AR model, respectively; $\Theta(B)$ is the polynomial of the MA model, $\Theta(B) = 1 - \sum_{j=1}^{q} \theta_j B^j$ , $q$, $\theta_j$ are the order and coefficient of the MA model respectively.

Modeling and predicted wind speed using the ARIMA model mainly includes the following three important steps:

1. Model order identification. The stationarity detection of the time series is carried out. The time series should be converted to a stationary time series using differential operation if a non-stationary series is detected. Then, the differential order $d$ can be determined. After that, the model orders $p$ and $q$ can be determined according to the AIC criteria by calculation of the Auto-Correlation function (ACF) and the Partial ACF (PACF).

2. Estimation of the model parameters. The maximum likelihood method is usually adopted to estimate the model parameters.
3. Diagnostic checking and prediction. Whether the model is suitable for the series is determined, and the future wind speeds are predicted by the constructed ARIMA model.

### 2.3. The Support Vector Machine (SVM)

The support vector machine was proposed by Vapnik et al. in 1995 [28], developed from the statistical learning theory and the structural risk minimization principle. The most notable feature of SVM is that it can effectively overcome the large deviation of the prediction results and problems such as over-learning, dimensional disaster, and local extremum. It is suitable for dealing with small samples, high dimensionality, and nonlinearity. The basic principle of the SVM regression is to map the data of the input space into a high-dimensional feature space through a nonlinear mapping, after which the linear regression is performed in this feature space. Suppose we are given training data $\{(x_1, y_1), (x_2, y_2), \cdots, (x_n, y_n)\}$, the SVM regression function is formulated as follows [29]

$$f(x) = \sum_{i=1}^{N} \omega_i \phi(x_i) + b \tag{6}$$

where $N$ denotes the number of the training data, $x_i$ is the input patterns, $y_i$ is the output patters; $\omega_i$ is the regression coefficients vector; $\phi(x_i)$ is the nonlinear mapping function in the feature space from the input patters $x_i$; $b$ is the bias term.

Actually, the solving of the regression function is converted to the quadratic programming problem. It gets [30,31]

$$\min\left\{\frac{1}{2}\|\omega\|^2 + C\sum_{i=1}^{N}\left(\xi_i + \xi_i^*\right)\right\} \tag{7}$$

$$\text{subject to } \begin{cases} y_i - \langle w, \phi(x_i)\rangle - b \le \varepsilon + \xi_i \\ \langle w, \phi(x_i)\rangle + b - y_i \le \varepsilon + \xi_i^* \\ \xi_i, \xi_i^* \ge 0 \ \ (i = 1, 2, \cdots, n) \end{cases} \tag{8}$$

where $\xi_i, \xi_i^*$ are the slack variables; $\varepsilon$ is the tolerance error between $f(x)$ and the output patters $y_i$; $C > 0$ is a constant that determines penalty of the samples exceeding the tolerance error.

The Lagrange function is constructed by introducing a dual set of variables in order to solve the quadratic programming problem. It deduces the dual optimization problem. It gets

$$\max\left\{\begin{matrix} -\dfrac{1}{2}\sum_{i=1}^{l}\sum_{j=1}^{l}\left(\alpha_i - \alpha_i^*\right)\left(\alpha_j - \alpha_j^*\right)\langle\phi(x_i), \phi(x_j)\rangle \\ -\varepsilon\sum_{i=1}^{l}\left(\alpha_i + \alpha_i^*\right) + \sum_{i=1}^{l} y_i\left(\alpha_i - \alpha_i^*\right) \end{matrix}\right\} \tag{9}$$

$$\text{subject to } \sum_{i=1}^{l}\left(\alpha_i - \alpha_i^*\right) = 0, \ \ \alpha_i, \alpha_i^* \in [0, C] \tag{10}$$

where $\alpha_i, \alpha_i^*$ are Lagrange multipliers; $\langle\cdot,\cdot\rangle$ denotes the dot product. Then, Equation (1) can be rewritten as follows

$$f(x) = \sum_{i=1}^{N}(\alpha_i - \alpha_i^*)\langle\phi(x_i),\phi(x_j)\rangle + b \tag{11}$$

$$w = \sum_{i=1}^{N}(\alpha_i - \alpha_i^*)\phi(x_i) \tag{12}$$

The kernel function is the essence of SVM. By means of the nonlinear mapping, the input patterns are expressed in the high-dimensional feature space, in which the linearly inseparable patterns in a low-dimensional space may be linearly separable, according to the pattern recognition theory. However, there are problems in directly classifying or regressing to determine the form and parameters of the nonlinear mapping function in the high-dimensional space. The biggest obstacle is the inner operation in the high-dimensional feature space, which can be effectively solved by introducing kernel function technology. The kernel function represents the inner product in the high-dimensional feature space:

$$K(x_i, x_j) = \langle\phi(x_i),\phi(x_j)\rangle \tag{13}$$

Then, Equation (4) can be expressed as follows:

$$\max\left\{ \begin{array}{l} -\dfrac{1}{2}\sum_{i=1}^{l}\sum_{j=1}^{l}(\alpha_i - \alpha_i^*)(\alpha_j - \alpha_j^*)K(x_i, x_j) \\ -\varepsilon\sum_{i=1}^{l}(\alpha_i + \alpha_i^*) + \sum_{i=1}^{l}y_i(\alpha_i - \alpha_i^*) \end{array} \right\} \tag{14}$$

$$\text{subject to } \sum_{i=1}^{l}(\alpha_i - \alpha_i^*) = 0, \ \alpha_i, \alpha_i^* \in [0, C] \tag{15}$$

where $K(x_i, x_j)$ is kernel function, any function satisfying Mercer's condition can be used as the kernel function. The Gaussian kernel function is adopted in this study:

$$K(x_i, x_j) = \exp\left[-\frac{\|x_i - x_j\|^2}{2\delta^2}\right] \tag{16}$$

Then, Equation (6) can be rewritten as follows

$$f(x) = \sum_{i=1}^{N}\omega_i\phi(x_i) + b = \sum_{i=1}^{N}(\alpha_i - \alpha_i^*)K(x_i, x_j) + b \tag{17}$$

## 3. Framework of the Proposed Hybrid Wind Speed Prediction Model

The decomposition effect of signals has been effectively improved by means of CEEMD, as it eliminates the problem of modal mixing in the process of signal decomposition. In addition, The ARIMA is very good at predicting linear data, and SVM has a good predictive effect on nonlinear data. Combining the advantages of the two single models, Pai and Lin et al. [32] propose an ARIMA-SVM hybrid prediction model, which is successfully applied to forecast the stock price. Therefore, a hybrid short-term wind speed forecasting model, combining the advantages of CEEMD, ARIMA, and SVM, is proposed. The framework of the proposed model is shown in Figure 1. First of all, The EMD or CEEMD are employed to decompose the original wind speed series into a variety of subsequences of IMFs and a residual component. Secondly, each of the IMF and the residual

are predicted by the ARIMA method to obtain the forecasting values. However, prediction errors are produced during the prediction process using the ARIMA model. To relieve the adverse influences of the errors on the prediction performance, a new subseries is reconstructed by superimposing all of the error subsequences. Thirdly, the SVM method is adopted to forecast the error subseries. Finally, all of the prediction subsequences are superimposed to obtain the predicted wind speed.

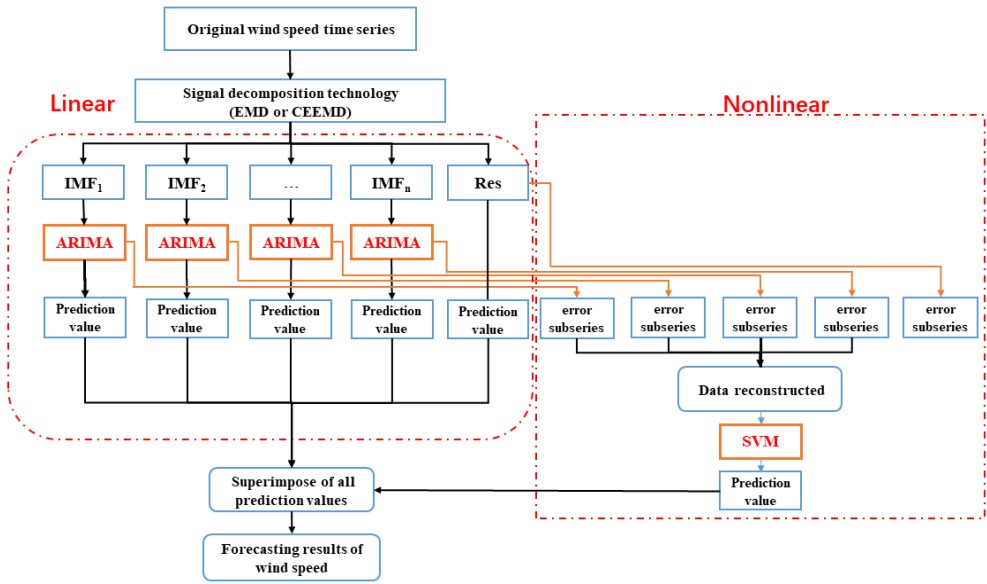

**Figure 1.** Framework of the hybrid forecasting model.

## 4. Experiments and Results Analysis

### 4.1. Description of Wind Speed Data

The research data comes from a field measured wind speed observation tower erected on south bank of the Qiongzhou Strait Bridge site. In order to verify the effectiveness and stability of the hybrid prediction model, typhoons data from typhoons Wutip and Ramason are selected as sample data. Ramason appeared in the Pacific Ocean, west of Guam, on the afternoon of 12 July 2014. Soon, it is upgraded to a strong typhoon and made landfall along the coast of Longtang Town, Xuwen County, Guangdong Province, China, on the 18th. The maximum wind speed was 60 m/s and the minimum pressure of the center was 950 hPa. Data collection lasted for 72 h, from 00:00 on 17 July 2014 to 24:00 on 19 July 2014. Tropical Storm Wutip formed at 14:00 on 27 September 2013 in the central South China Sea. On the 29th, it was upgraded to a strong typhoon in the waters of Sansha City, Hainan Province, and made landfall in Quang Binh Province, Vietnam on the 30th. The maximum wind speed was 35 m/s and the minimum pressure of the center was 970 hPa. Data collection lasted for 48 h from 00:00 on 29 September 2013 to 24:00 on 30 September 2013.

The time intervals of each step of the original wind speed data is 10 min. Dataset 1 comes from Typhoon Wutip. Sample data of Typhoon Ramason is divided into two datasets, i.e., dataset 2 and dataset 3, according to the variation of the mean wind speed. The wind speed fluctuation range of dataset 2 changes dramatically, while the wind speed of dataset 3 varies smoothly. The statistics of the three datasets are shown in Table 1, and the time-history of the datasets is depicted in Figure 2.

<p style="text-align:center"><strong>Table 1.</strong> Statistic features of datasets</p>

| Typhoon | Sample Data | Number of Data | Training Set | Test Set | Mean Wind Speed (m/s) | Range of Wind Speed (m/s) | Volatility Level |
|---|---|---|---|---|---|---|---|
| Wutip | Dataset 1 | 288 | 200 | 88 | 10.23 | 3.64~21.69 | moderate |
| Ramason | Dataset 2 | 432 | 300 | 142 | 11.44 | 0.29~47.12 | high |
| | Dataset 3 | 180 | 100 | 80 | 4.18 | 0.3~8.47 m/s | low |

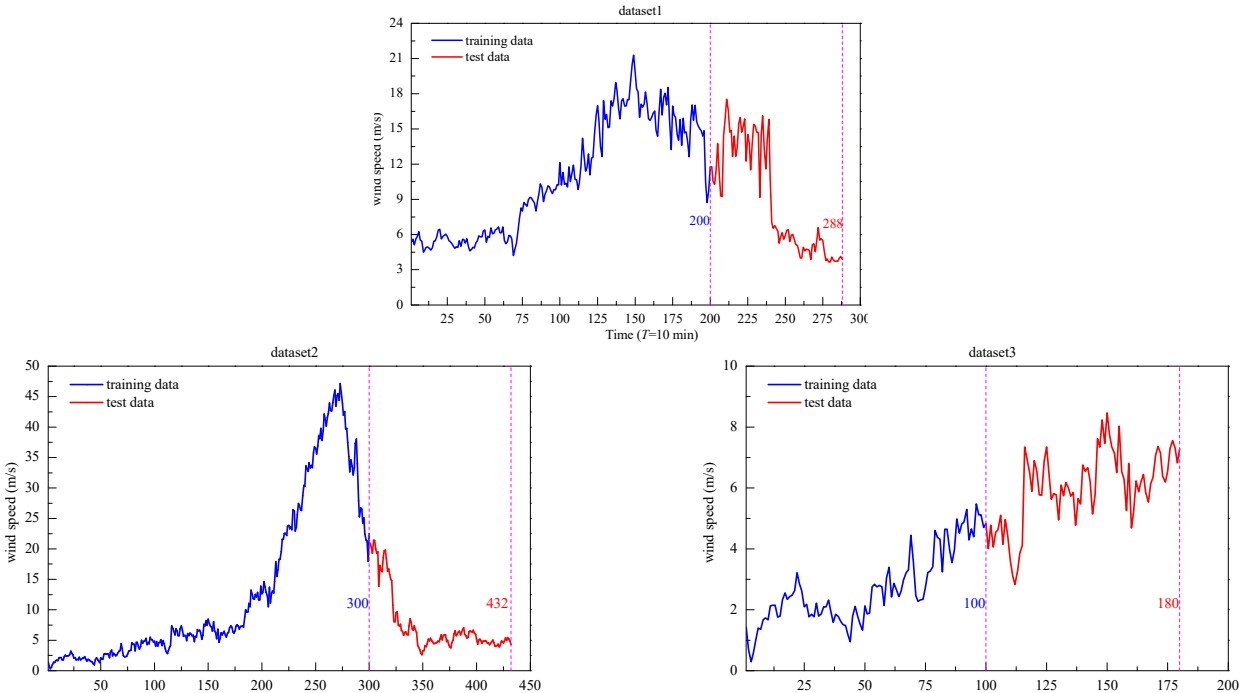

**Figure 2.** Three wind speed datasets.

### 4.2. Evaluation Indexes

To measure the prediction performance of the wind speed prediction model, three evaluation indexes, including *MAE*, *MAPE*, and *RMSE* [13,33], are selected to verify the prediction accuracy of the model proposed in this paper. The evaluation indexes are as follows:

1. Mean absolute error

$$MAE \; = \; \frac{1}{N} \sum_{t=1}^{N} \left| Z(t) \, - \, Z'(t) \right| \tag{18}$$

2. Mean absolute percentage error

$$MAPE = \frac{100\%}{N} \sum_{t=1}^{N} \left| \frac{Z(t) - Z'(t)}{Z(t)} \right| \tag{19}$$

3. Root mean squared error

$$RMSE \; = \; \sqrt{\frac{1}{N} \sum_{t=1}^{N} (Z(t) \, - \, Z'(t))^2} \tag{20}$$

where $Z(t)$ is the measured wind speed at a certain moment in time; $Z'(t)$ is the predicted wind speed; and *N* is the forecasted wind speed. The smaller the value of the three evaluation indexes, the higher the prediction accuracy of the forecasting model.

### 4.3. Analysis of Comparative Results

The accuracy of each step for the hybrid model has a significant impact on the final prediction results. To illustrate the modeling process of the wind speed prediction model proposed in this paper, a detailed demonstration of the modeling process is presented in this section, taking dataset 1 as an example.

Firstly, the signal decomposition algorithms of EMD and CEEMD are introduced to decompose the original wind speed into a series of IMFs and a residual component, as shown in Figure 3. The raw wind speed presents a large fluctuation range, while volatility of the decomposed IMF components behave much more smoothly. With the decrease in the fluctuation scales, the IMF series gradually tends to be stationary. A total of seven IMF subseries are searched by CEEMD, while EMD only determine four IMF subseries. Generally, the decomposition process may be affected by the volatility level of the wind speed series, to some extent. More IMFs can be obtained for wind datasets with larger fluctuations and complex frequency components. Owing to the adding of opposing white noise to the original wind speed, the CEEMD effectively avoids the disadvantages of modal mixing of EMD, which is a more ongoing decomposition method. As the CEEMD algorithm has a higher decomposition precision, it is easy for the CEEMD algorithm to determine more IMF components with different scales.

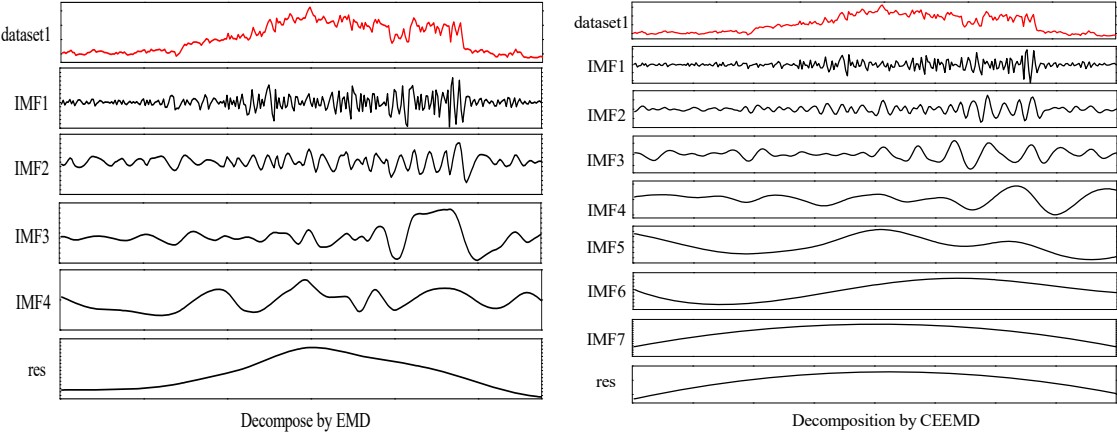

**Figure 3.** EMD and CEEMD decomposition diagram of dataset 1.

Then, the ARIMA model is adopted to rebuild the IMF subseries, and the forecasting wind speed subseries are available, taking advantage of the established ARIMA model. The prediction results are depicted in Figure 4. The model presents good prediction effects for the low-frequency IMF components, as the ARIMA has a strong capturability of the volatility of the wind speed subseries with low-frequency. It shows that ARIMA models have sufficient prediction accuracy for low-frequency components. However, a certain prediction error exists for the high-frequency components. However, due to the linear features of ARIMA, the prediction model is inevitably subject to prediction error for the forecasting of the high-frequency components.

Further, the prediction errors of all components are superimposed to reconstruct a new error subseries. Given that nonlinear features of the error subseries, the SVM are employed to predict the error results, as shown in Figure 5. In spite of the complex volatility of the error subseries, the SVM model fits the prediction value close to the true value, to some extent, which displays the great power of the nonlinear processing ability. It is feasible to apply the SVM model to forecast the error subseries.

Finally, the prediction results are obtained by superimposing all of the prediction values gained from the above steps.

To further verify the stability and accuracy of the model proposed in this paper, a comparative analysis is conducted between the model and five other representative prediction modes, namely ARIMA, SVM, ARIMA-SVM, EMD-ARIMA, and CEEMD-ARIMA. The comparative results are depicted in Figures 6–8, which show the deviations between the raw wind speeds and the forecasted wind speeds according to the prediction models. The proposed models show good performances; the prediction values are much closer to the true wind speed than other prediction models. Table 2 lists the quantitative evaluation index results using the prediction models for the three datasets.

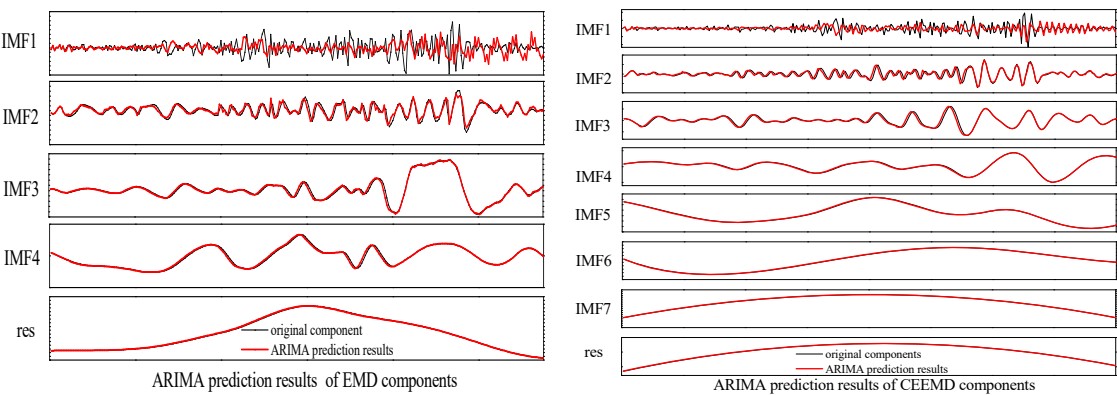

**Figure 4.** ARIMA prediction results.

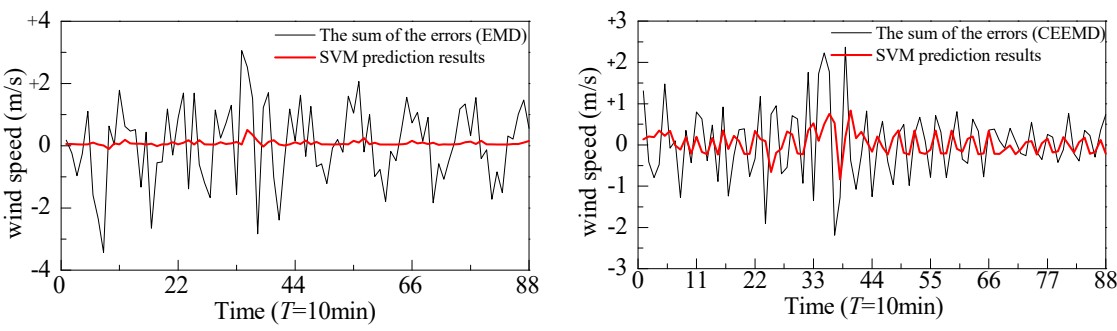

**Figure 5.** SVM prediction results.

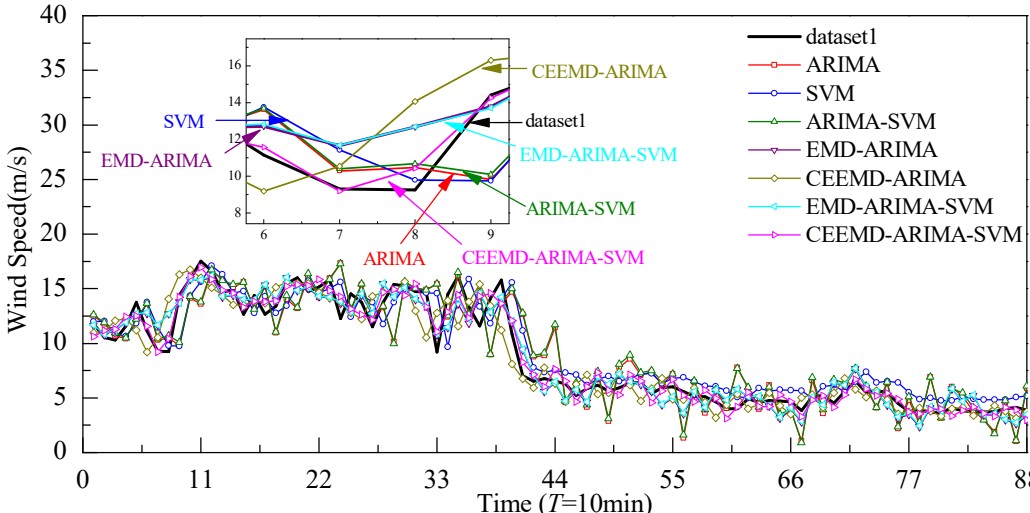

**Figure 6.** Comparative prediction results for dataset 1.

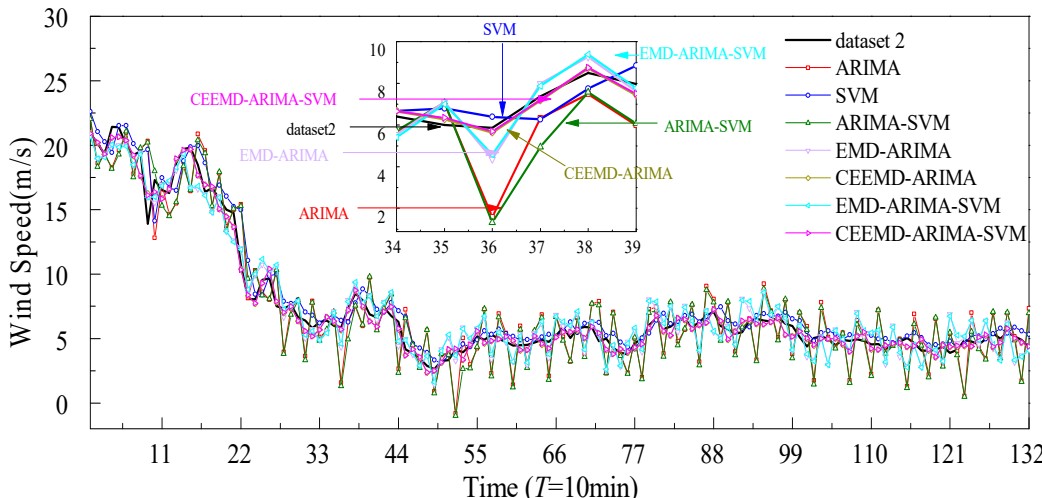

**Figure 7.** Comparative prediction results for dataset 2.

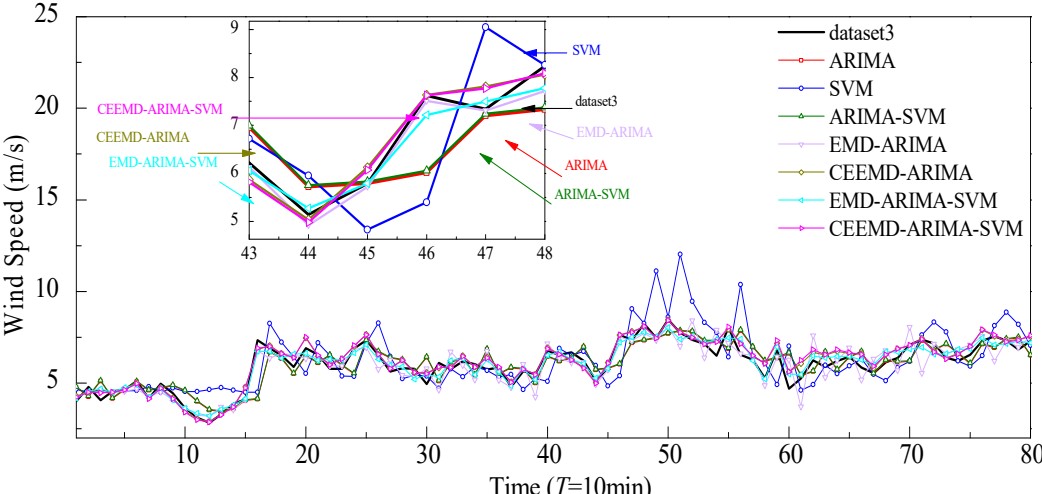

**Figure 8.** Comparative prediction results for dataset 3.

**Table 2.** Evaluation index results for the prediction models

| | Dataset 1 | | | Dataset 2 | | | Dataset 3 | | |
|---|---|---|---|---|---|---|---|---|---|
| | **MAE (m/s)** | **MAPE (%)** | **RMSE (m/s)** | **MAE (m/s)** | **MAPE (%)** | **RMSE (m/s)** | **MAE (m/s)** | **MAPE (%)** | **RMSE (m/s)** |
| ARIMA | 1.787 | 25.122 | 2.269 | 1.747 | 29.222 | 2.116 | 0.557 | 9.768 | 0.751 |
| SVM | 1.391 | 18.777 | 1.748 | 0.746 | 11.226 | 1.091 | 0.940 | 15.959 | 1.267 |
| ARIMA-SVM | 1.821 | 25.851 | 2.283 | 1.767 | 29.627 | 2.116 | 0.556 | 9.829 | 0.753 |
| EMD-ARIMA | 1.037 | 14.178 | 1.271 | 1.036 | 17.708 | 1.264 | 0.498 | 8.381 | 0.655 |
| CEEMD-ARIMA | 0.669 | 8.046 | 0.849 | 0.417 | 6.710 | 0.537 | 0.289 | 5.026 | 0.375 |
| EMD-ARIMA-SVM | 1.027 | 14.075 | 1.260 | 1.032 | 17.548 | 1.245 | 0.477 | 8.103 | 0.628 |
| CEEMD-ARIMA-SVM | 0.664 | 8.098 | 0.839 | 0.412 | 6.672 | 0.529 | 0.288 | 5.010 | 0.377 |

It is clearly seen from the figures and Table 2 that the evaluation indexes for the single ARIMA and SVM models are much more accurate than those for the other hybrid models, which indicates the single ARIMA and SVM prediction models have low accuracy. Moreover, the prediction wind speed presents the phenomenon of hysteresis. The possible



causes of the results is that the single model cannot identify and update the model parameters in time, due to the sudden changes in the wind speed. In turn, it further verifies the feasibility and efficiency of the hybrid models.

It is obvious that the decomposition technology has greatly improved the prediction accuracy, as seen by comparing the prediction results of EMD/CEEMD-ARIMA-SVM model with those of the ARIMA-SVM model. For dataset 1, the three evaluation indexes of MAE, MAPE, and RMSE for the EMD-ARIMA-SVM model are 1.027, 14.075, and 1.260, respectively, which are 43.6%, 45.6%, and 44.8% lower than those of the ARIMA-SVM model. Meanwhile, the indexes of MAE, MAPE, and RMSE for the CEEMD-ARIMA-SVM model are 0.664, 8.098, and 0.839, respectively, which are 63.5%, 68.7%, and 63.2% lower than those of the ARIMA-SVM model. Moreover, since a thoroughly decomposition of the original data is carried out by CEEMD, the prediction performance of the CEEMD-ARIMA-SVM is better than that of the EMD-ARIMA-SVM model. A similar conclusion can be found for the hybrid model applied to dataset 2 and dataset 3. Comparatively speaking, due to the greatest volatility of dataset 2, the evaluation indexes of the hybrid model CEEMD-ARIMA-SVM have the greatest reduction in effectiveness, reaching an astonishing level of a greater than 70% reduction. For relatively stationary dataset 3, the evaluation indexes of the hybrid model CEEMD-ARIMA-SVM are also reduced by nearly 50%.

Generally speaking, the prediction performance of the EMD/CEEMD-ARIMA-SVM model is slightly better than the EMD/CEEMD-ARIMA model. However, the evaluation indexes have not improved significantly. The reason may be that the subseries is stationary and linear after the decomposition of EMD or CEEMD, and the prediction of the subseries by ARIMA has achieved good prediction results. The error subseries is so small that the error subseries prediction results of the SVM have little effect on the overall prediction results.

## 5. Conclusions

Improving the wind speed prediction accuracy is of great significance for the operation of the wind monitoring and warning system, which is involved in the running safety of vehicles under crosswinds. In this paper, a novel short-term wind speed prediction model, combined with EMD/CEEMD, ARIMA, and SVM, is proposed for the application of forecasting typhoon wind speed. In the proposed EMD/CEEMD-ARIMA-SVM model, the EMD or CEEMD is adopted to decompose the original wind speed into a series of subsequences. The ARIMA is used to predict the wind speed of the subsequences. Finally, the prediction errors of all of the subsequences are reconstructed and predicted by the SVM algorithm. The efficiency and accuracy of the model are verified by three wind speed sample datasets, in comparison with the others wind prediction models. The following conclusions can be drawn:

1. The hybrid model shows higher prediction accuracy than the single model. The hybrid model is more suitable for higher volatility of wind speeds, exhibiting the ability to capture the fluctuating characteristics of wind speeds, while the single ARIMA model is more suitable for less volatile data.
2. The EMD and CEEMD can reduced the nonstationarity and nonlinearity of the original wind speed. It decomposes the raw wind speeds into a series of subsequences, greatly reducing wind speed volatility. The prediction accuracy of the hybrid models has been obviously improved with the aid of the decomposition technologies, such as EMD and CEEMD, since CEEMD has removed the disadvantage of the appearance of modal mixing for EMD. Overall, the prediction performance of the CEEMD-ARIMA-SVM is better than that of the EMD-ARIMA-SVM model.
3. Taking the three wind speed datasets as experiment examples, the prediction performance of the proposed EMD/CEEMD-ARIMA-SVM wind prediction model achieved

optimum results according to the minimal evaluation indexes of MAE, MAPE, and RMSE.

4.  It seems that the prediction performance of the hybrid model mainly relies on the combination of CEEMD with ARIMA. The SVM method has only slight effects on the prediction performances of the hybrid model, as the error subseries prediction results using the SVM method show few improvement effects on the overall prediction results.

In a summary, the hybrid wind speed prediction model proposed in this paper is a feasible wind speed forecasting algorithm. It has sufficient accuracy for typhoon wind speed prediction. The method can be treated as an alternative wind speed prediction method, which can also be applied to other wind speed prediction scenarios, such as the wind power prediction for wind farms, the migration of pollutants in the fields of environmental protection, and so on.

**Author Contributions:** The experiment, data analysis, and writing of the paper were conducted by N.C.; the experiment and data analysis were completed by H.S. and Q.Z.; validation, methodology, and paper editing were handled by S.L. and Q.Z. All authors have read and agreed to the published version of the manuscript.

**Funding:** This work is supported by NSFC (Grant Nos. 51778228, 52078210) and the National Science Fund for Distinguished Young Scholars of Hunan Province (Grant No. 2021JJ10003).

**Institutional Review Board Statement:** Not applicable.

**Informed Consent Statement:** Not applicable.

**Data Availability Statement:** No new data were created or analyzed in this study. Data sharing is not applicable to this article.

**Acknowledgments:** We sincerely appreciate Wang Xiuyong for giving supervision and guidance in the writing process.

**Conflicts of Interest:** The authors declare no conflict of interest.

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
