# Peer review of "A Short-Term Wind Speed Forecasting Model Based on EMD/CEEMD and ARIMA-SVM Algorithms"

_applsci, doi:10.3390/app12126085_

Round 1

Reviewer 1 Report

Please, see attached document.

Reviewer 2 Report

In this paper, the authors proposed a short-term wind speed forecasting model, combining with complementary ensemble empirical mode decomposition (CEEMD), auto-regressive integrated moving average (ARIMA) and support vector machine (SVM) technology. This subject is important in the area of security and safety in windy environments.

I have the following comments and recommendations

1.      The title: I suggest removing the word “Novel”, let the readers perceive the novelty in the manuscript.

2.       In the abstract: are the mentioned RMSE 0.839, 0.529 and 0.377 in %  (see also table 2)?

3.      As the authors said between line 55 and line 57: …… Techniques such as EMD …. WPT (Wavelet Packet Transform) [10] ….. I suggest comparing results obtained when decomposing the wind time series by CEEMD and then by wavelet decomposition (see Figure 1. Framework of the hybrid forecasting model).

4.      Are the statistical characteristics of wind time series affected by the decomposition process?

5.      In the EMD transform the decomposition technique is nonlinear but a linear sum of IMFs reconstructs the original signal. Also, EMD decompose the signal from the highly frequency IMF to the Lower frequency IMF. How much this procedure of the decomposition affects the results of forecasting if we use a variety of wind time series with different speeds and frequencies? If we replace the EMD decomposition by wavelet decomposition, what will be the results and the performance parameters (RMSE, MAPE …)?

6.      How we can apply this methodology in real world application? Is CEEMD decomposition mathematically completed to implement it in such application?

7.      Limits of this study should be discussed in the conclusion.

Round 2

Reviewer 1 Report

All the comments have been addressed. I believe that the manuscript is ready for publication. I am adding few very minor suggestions that I hope could help to further improve the manuscript:

1) L. 98-105: I would consider to remove the description in the Introduction and refer to the new description provided in Section 3 for the details of the proposed model. The two provided descriptions seem a bit repetitive.

2) L. 170-174 : Please, be aware that there are still few terms left which are not in italic (d, p, q).

Reviewer 2 Report

 As a research paper, the importance of this work lies in the minimization of risks in windy periods. I hope that the authors will replace the empirical modal decomposition by the wavelet decomposition and its variants to see the difference and also improve the performance of the prediction in a future work. As they stated in answer #3:

“…………Since there is no relevant research foundation, it is difficult to complete the relevant work in a short period of time. According to your suggestions, the research on the decomposition techniques such as wavelet decomposition and wavelet packet transformation would be conducted in the follow-up research work. It is meanful to investigate effects of the wavelet decomposition and WPT on accurcy of the wind speed forecasting model. ”

Decision:  Accept, I have no other concerns.
